DCEMRI.jl: a fast, validated, open source toolkit for dynamic contrast enhanced MRI analysis

Smith David S. 1 2 david.smith@vanderbilt.edu
Li Xia 1 2
Arlinghaus Lori R. 1 2
Yankeelov Thomas E. 1 2 4 5
Welch E. Brian 1 2 3
1 Institute of Imaging Science, Vanderbilt University , Nashville, TN , USA
2 Department of Radiology and Radiological Sciences, Vanderbilt University , USA
3 Department of Biomedical Engineering, Vanderbilt University , USA
4 Department of Physics and Astronomy, Vanderbilt University , USA
5 Department of Cancer Biology, Vanderbilt University , USA
Frauscher Ferdinand
Electronic publication date: 2015 Apr 23
Publication date: 2015
Volume: 3
Electronic Location ID: e909
Received 2014 Dec 5; Accepted 2015 Apr 3
Copyright: © 2015 Smith et al.
Copyright year: 2015
Copyright holder: Smith et al.
License: This is an open access article distributed under the terms of the Creative Commons Attribution License, which permits unrestricted use, distribution, reproduction and adaptation in any medium and for any purpose provided that it is properly attributed. For attribution, the original author(s), title, publication source (PeerJ) and either DOI or URL of the article must be cited.
License URL: https://creativecommons.org/licenses/by/4.0/

Keywords: Magnetic resonance imaging, DCE, Quantitative imaging biomarkers, qMRI, Cancer, Parallel computing, Julia, Medical imaging, Numerical methods

Funding: National Cancer Institute of the National Institutes of Health K25CA176219, U01CA142565, R01CA129961, R25CA092043 This project was funded by the National Cancer Institute of the National Institutes of Health, under Award Numbers K25CA176219, U01CA142565, R01CA129961, R25CA092043. The funders had no role in study design, data collection and analysis, decision to publish, or preparation of the manuscript.

==============================
We present a fast, validated, open-source toolkit for processing dynamic contrast enhanced magnetic resonance imaging (DCE-MRI) data. We validate it against the Quantitative Imaging Biomarkers Alliance (QIBA) Standard and Extended Tofts-Kety phantoms and find near perfect recovery in the absence of noise, with an estimated 10–20× speedup in run time compared to existing tools. To explain the observed trends in the fitting errors, we present an argument about the conditioning of the Jacobian in the limit of small and large parameter values. We also demonstrate its use on an in vivo data set to measure performance on a realistic application. For a 192 × 192 breast image, we achieved run times of <1 s. Finally, we analyze run times scaling with problem size and find that the run time per voxel scales as O(N1.9), where N is the number of time points in the tissue concentration curve. DCEMRI.jl was much faster than any other analysis package tested and produced comparable accuracy, even in the presence of noise.

Introduction

Dynamic contrast enhanced MRI

Dynamic contrast enhanced magnetic resonance imaging (DCE-MRI) involves the continuous acquisition of heavily T1-weighted MR images while a paramagnetic contrast agent (CA) is injected. The CA increases the contrast between different tissues by changing their inherent relaxation rates. By collecting serial images, each image voxel yields an intensity time course that can be used to estimate physiological parameters, such as the volume transfer constant Ktrans, extravascular extracellular volume fraction ve, and the plasma volume fraction vp (Choyke, Dwyer & Knopp, 2003; Yankeelov & Gore, 2009). Due to this, DCE-MRI has successfully been applied to assess vascular characteristics in both pre-clinical (Zwick et al., 2009; Jensen et al., 2010) and clinical settings (Lockhart et al., 2010; Mannelli et al., 2010).

The MR scanner typically handles the reconstruction of the acquired raw MR data into images, while the second step of DCE-MRI analysis is left to the end user. This second step includes determining a subset of voxels to process, fitting a nonlinear signal model to the time curve of each of those voxels, postprocessing the fitted model parameters, and summarizing the results.

Existing analysis software

In both clinical and research settings, a rapid, validated DCE-MRI analysis software package is a useful tool in the growing area of quantitative MR imaging. Furthermore, making a software package open source can increase the safety of clinical medical devices through community auditing, bug tracking, and version control.

Several DCE-MRI analysis packages have been released to the community. DCE@urLAB1 from Ortuño et al. (2013) has been validated against reference phantoms and includes many pefusion models, but it requires IDL, a commercial software package. It can be run for free with the IDL Virtual Machine, which requires registration and approval from the vendor. In the end the full IDL development environment must be installed, even though its functionality is crippled without a paid license. Second, DCE@urLAB was built around a graphical user interface (GUI) that did not provide batch processing. It also did not run on Mac OS X or Linux. Finally, the stated run times (Ortuño et al., 2013) were slower than we expected, given the computational complexity of nonlinear least squares fitting, suggesting some inefficiency due to the software complexity (∼20,000 lines of IDL code) or GUI overhead.

Zöllner et al. (2013) built an OsiriX plug-in called UMMPerfusion.2 This is a validated, open-source software package for DCE-MRI analysis, but it works only within the OsiriX image viewer, a Mac OS X only program and which only a 32-bit, basic version was available for free. This package also did not provide a headless batch processing tool, and the reported run times were also slower than expected, even though it supported parallel processing.

The DCE Tool3 is another GUI solution for DCE modeling. It requires both a Windows only software package called ClearCanvas (available for free) and the MATLAB4 run time environment (a commercial product) in order to run. This solution was quite complex, comprising over 1.2 million lines of C and C# code.

The most useful existing tool for our needs was dcemriS45 (Schmid et al., 2009a; Schmid et al., 2009b; Schmid et al., 2006), an R package. R is a popular statistical analysis language, but it is not as common in MRI research. Nevertheless, the package contained many advanced models, was fast in our benchmarks, and included parallel processing, but it still took tens of seconds to process a typical breast DCE-MRI data set and to the best of our knowledge has not been validated.

Why Julia?

We desired a fast, free, simple, easily extendable code that required a minimal installation, would work on Windows, Mac OS X, and Linux, and would be familiar to MATLAB users. Based on these criteria, we chose Julia as the implementation language. Julia (Bezanson et al., 2012; Bezanson, Edelman & Karpinski, 2014) is a new, high-level language designed for technical computing and that approaches the performance of C and Fortran.6 It contains extensive libraries for linear algebra and signal processing and provides distributed parallel execution. It also easily interoperates with existing scientific languages, such as C, Fortran, and Python, making it an excellent glue language for scientific computing. Interested readers can get a flavor of Julia’s simultaneous efficiency and simplicity with the example in Listing 1.

In short, Julia feels like MATLAB, which is simple and familiar to many investigators, but runs faster and is completely free. In particular for DCE-MRI, Julia’s simple and flexible parallel computing model allows excellent parallelization of the nonlinear least squares fitting problem.

Goals

We developed DCEMRI.jl with five features in mind: open source, free, fast, flexible, and simple. Open source enables auditing, bug finding, and community improvement. Free software reduces the barrier to use and adoption. A faster package saves investigator time and is more clinically practical. Flexibility anticipates new uses and heterogeneous adoption. And simple code design reduces bugs through easier auditing and yields greater didactic value. Here we present the results of the effort to produce such a package.

Materials and Methods

Units

All input variables are assumed to be in SI units, except flip angles which should be given in degrees. All output is in SI units except for Ktrans which is scaled to units of min−1 to maintain convention within the DCE-MRI community.

Listing 1 The function f(x, y) is compiled upon first execution using Julia’s just-in-time (JIT) compiler, making subsequent function calls much faster. The generated CPU instructions can be viewed directly with the code_native function, which takes two arguments: a function name and an n-tuple of argument types. Here we examine the native CPU code produced if f were called on two 64-bit floating point variables. The native code contains only one math instruction (vaddsd), surrounded by the code required to preserve and restore the stack for a function call. (By convention the first two floating point arguments are passed into the function in the XMM0 and XMM1 registers, and the result is placed in XMM0.) All other overhead has been stripped away automatically without an explicit compilation. The development and testing cycle is accelerated by this elimination of a separate code compilation step, and the generated code executes as quickly as compiled C or Fortran would.

julia> f(x,y) = x + y f (generic function with 1 method) julia> f(2,3) 5 julia> code_native(f, (Float64, Float64)) .section __TEXT,__text,regular,pure_instructions push RBP mov RBP, RSP vaddsd XMM0, XMM0, XMM1 pop RBP ret

Reproducible research

DCEMRI.jl is maintained under version control in a GitHub7 archive. The exact version of the code used to produce the results here may be obtained by “pinning” module at version 0.1.0 using the command Pkg.pin("DCEMRI", v"0.1.0") in the Julia shell. This will effectively reverse any code changes committed to the repository after the publication of this paper. The command Pkg.free("DCEMRI") will “unpin” and get the latest updates again.

Bugs and suggestions can be filed by users at the GitHub repository by filing “issues.” Each issue is tracked and can be connected to subsequent code patches so that every change to the code base can be traced in case of a breaking change or feature addition. Additionally, each and every line can be traced back to the user that last changed it using GitHub’s “blame” system.

Input parameters

For simplicity and maximum compatibility, DCEMRI.jl reads and writes input as Matlab MAT v5 files. This allows users to call DCEMRI.jl from any language that can read and write MAT files. This list includes MATLAB, Octave, Python, and R. The input MAT file must include

1. vector t of imaging time points,

2. vector Cp of the arterial input function,

3. n-D array DCEdata of dynamic data with time in the first dimension, and

4. either a map of the pre-contrast R1 relaxation rate R1(x, 0) and associated signal S(x, 0) (R10, S0) or a series of T1-weighted multiflip data and associated flip angles (T1data, T1flip).

Note that for data with unreliable flip angle information, a separately computed R1 and S0 map is preferred for the best accuracy. All other parameters are optional and will be supplied with defaults if not provided. The defaults may be overridden by supplying additional parameters in the MAT file, command-line arguments, or function parameters.

T1 mapping

DCEMRI.jl can accept as input an R1 longitudinal relaxation rate map (R1 ≡ 1/T1) or multiple flip angle (multiflip) T1-weighted dynamic data. If the multiflip data is supplied, the code will fit R1 and signal density maps using the signal equation for a spoiled gradient echo sequence: (1) Sx,t=Sx,0sin θ1−exp−R1x,tTR1−cos θexp−R1x,tTR,

where S(x, t) is the signal as a function of space x and time t, θ is the flip angle, and TR is the repetition time. Here we have ignored R2∗ decay because we are assuming that the echo time TE is much shorter than 1/R2∗.

If an R1 fit is required, only voxels with a mean signal intensity of at least 10% of the maximum intensity are fit, to avoid fitting voxels that are dominated by noise. This cutoff was chosen empirically and can be customized to the signal distribution in a given data set. Voxels selected for fitting are then split evenly across CPU cores for Levenberg–Marquardt fitting to the signal equation. The results for each voxel subset are returned to the parent process, where full maps of R1(x, 0) and S(x, 0) are formed. The fact that fitting of individual voxels is independent of neighboring voxels is crucial to allowing this problem to be efficiently parallelized. All models implemented so far in DCEMRI.jl take voxels to be independent of their neighbors.

DCE fitting

To fit a model to the supplied DCE data, the raw MR signal is converted first to an effective R1(x, t) relaxation rate by inverting the signal equation (Eq. (1)): (2) R1x,t=−1TRlog1−sx,t+sx,texp−R1x,0TR−exp−R1x,0TRcos θ1−sx,tcos θ+sx,texp−R1x,0TRcos θ−exp−R1x,0TRcos θ,

where s(x, t) = S(x, t)/S(x, 0) is the signal normalized at time t = 0, θ is the flip angle (assumed constant), and TR is the repetition time. Note that we have eliminated all terms involving sin θ. Since the error in sin θ is larger than the error in cos θ when θ is close to zero, eliminating sin θ reduces sensitivity to inhomogeneities in the volume excitation (also known as B1 transmit radio frequency field inhomogeneities). If a B1 field map is available, it can be used to generate an R1 map separately that can be used as an input to DCEMRI.jl. Currently DCEMRI.jl does not support R1 mapping with spatially varying flip angles, although nothing precludes adding that functionality in the future.

In the next stage of the processing, the effective relaxation rate R1(x, t) is converted to the concentration in tissue Ct of the contrast agent using Ctx,t=R1x,t−R1x,0r1,

where r1 is the relaxivity of the contrast agent. For our in vivo experiment, Gd-DTPA was used, for which we take the relaxivity to be 4.5 s−1 mM−1 at 3.0 T because that was the relaxivity used in the validation data, and this same value has been found in in vivo studies Sasaki et al. (2005). This value can also be specified by the user.

Next an optional mask of voxels to process can be supplied in the MAT file as the variable mask. If not supplied, an automatic mask is generated from a variation of the signal enhancement ratio (SER, Hylton et al., 2012), defined here as the mean signal in each voxel in the last three dynamics divided by the mean of the signal in the voxel in the first three dynamics. (Note that this requires the acquisition of three pre-contrast time points.) By default any voxels with an SER above 2.0 will be included in the processing mask. This cutoff can be changed by the user.

Tissue models

Three main tissues models are included by default: the Standard and Extended Tofts-Kety models (Yankeelov & Gore, 2009) and a plasma-only model (no exchange limit). Other models can be added easily by the user. The Extended Tofts-Kety model is a two-compartment model that assumes that the blood vessel supplies the CA to the tissue at a slow and fixed transport rate Ktrans. The volume of the extracellular, extravascular tissue space is labeled ve, and the volume fraction of the blood vessels is vp. Under this model, the tissue concentration can be written as (3) Ctx,t=Ktransx∫0tCpsexpkepxs−t ds+vpx Cpt,

where the efflux rate constant (4) kepx≡Ktransxvex.

Fitting the derived tissue concentration curves Ct to this model involves finding the Ktrans, ve, and vp that best reproduce the observed Ct given an AIF Cp(t). In the Standard Tofts-Kety model, vp is assumed to be zero, and in the plasma-only model Ktrans is assumed to be zero. Formulating the integral using kep instead of the ratio Ktrans/ve produces better fits. To get ve, Ktrans can be divided by kep, taking care to handle cases where kep = 0.

The models to use are specified with a bitmask supplied in either the input MAT file or as a command line argument, and multiple models can be fit to the same voxel. The code will then choose the best fitting model based on the reduced χ2. For each model selected, numerical integration is performed using a trapezoidal rule, and the nonlinear least squares fitting is performed in parallel for each voxel independently using the Levenberg–Marquardt method. All fitting code is written in pure Julia—no external libraries are called.

Postprocessing

Parameters were clamped in voxels that where the fit produced unphysical values. The volume fractions ve and vp were clamped to the [0, 1] interval, while Ktrans was clamped to the [0, 5] interval. The original fit residuals were retained for filtering as well. If a fit residual is large, one can safely assume that either the signal in the voxel was too low to provide an accurate fit or the model assumptions were violated at that location. In either case, poorly fitted voxels should be omitted in an imaging analysis. The user should select the correct, data-dependent cutoff residual, so DCEMRI.jl does not automatically filter by residual.

Finally, all results are saved to an output MAT v5 file. The name of this file can be customized through a command line argument or the variable outfile in the input MAT file.

Note that DCEMRI.jl does not implement PACS archival, because the results are stored as MAT files. Additional software is needed to convert the MAT file to DICOM format for PACS archival. If a record of the required patient, exam and image metadata is maintained, it is theoretically straightforward to implement a conversion to DICOM format for PACS archival. However, support for that conversion is beyond the scope of this work.

QIBA phantom data

The Quantitative Imaging Biomarkers Alliance8 has provided virtual DCE phantoms in the DICOM format for validating DCE-MRI analysis codes. Several phantoms are available for benchmarking both DCE model fitting and T1 mapping, with a range of noise and timing errors added. Here we chose the noise-free Standard and Extended Tofts phantoms (versions 6 and 4, respectively). The Standard Tofts phantom contains 10 × 10squares of all combinations of six values of Ktrans ∈ {0.01, 0.02, 0.05, 0.1, 0.2, 0.35} min−1 and five values of ve ∈ {0.01, 0.05, 0.1, 0.2, 0.5}, for 30 regions total. The phantom contains 1,361 time points for each voxel. The Extended Tofts-Kety phantom (version 4) contains 10 × 10patches of all combinations of the parameters Ktrans ∈ {0.0, 0.01, 0.02, 0.05, 0.1, 0.2} min−1, ve ∈ {0.1, 0.2, 0.5}, and vp ∈ {0.001, 0.005, 0.01, 0.02, 0.05, 0.1}, for 108 regions total. This phantom contains 661 time points for each voxel. Figure 1 shows an example dynamic from the version 6 QIBA phantom along with its associated AIF.

Figure 1 Example validation data.

(A) The noise-free v6 QIBA phantom contains a numeric label of its order in the time series in the upper left, the tissues regions in the middle, and a vessel strip at the bottom, from which the AIF may be extracted. (B) The AIF extracted from this phantom. (A) The noise-free v6 QIBA phantom contains a numeric label of its order in the time series in the upper left, the tissues regions in the middle, and a vessel strip at the bottom, from which the AIF may be extracted. (B) The AIF extracted from this phantom.

In the noisy cases, we followed Ortuño et al. (2013) and added complex Gaussian noise with standard deviation σ = 0.2 relative to the pre-contrast baseline to the images. We then went a step further and took the magnitude of the resulting data, transforming the noise distribution into the more realistic Rician distribution. The difference between a Gaussian and a Rician distribution is minimal for voxels with signal-to-noise ratios ≳10. No noise was added to the AIF for simplicity and to allow faithful comparisons between this work and Ortuño et al. (2013).

We extracted just one voxel from each region in the noise-free cases to reduce the computation time, since all voxels were identical in each region. We retained all 100 voxels in each region for the noisy cases, however, in order to sample the effects of the added noise.

In vivo data collection

In vivo breast data were acquired using a Philips9 Achieva 3.0 T MR scanner. The scan protocol was optimized for use with the quantitative modeling in an IRB-approved10 ongoing clinical trial of response to neoadjuvant chemotherapy (Li et al., 2014), so it used a higher temporal resolution and lower spatial resolution than clinical protocols.

For the T1-weighted data, a 3D gradient echo multiple flip angle approach was used with TR = 7.9 ms, TE = 1.3 ms, and flip angles of 2–20 deg in 2 deg increments. Flip angles were uniformly spaced instead of optimized because of the broad range of tissue properties found in tumors. The acquisition matrix was 192 × 192 × 20 (full breast) over a sagittally oriented field-of-view of 22 cm × 22 cm × 10 cm. Scan time was just under 3 min. The DCE sequence used identical parameters but with a single flip angle of 20 deg. Each 20-slice set was collected in 16.5 s at 25 time points for approximately 7 min of scanning. A catheter placed within an antecubital vein delivered 0.1 mmol kg−1 of the contrast agent Magnevist at 2 mL s−1 (followed by a saline flush) via a power injector after the acquisition of three baseline dynamic scans for the DCE study. A population AIF was used Li et al. (2011).

Modes of operation

Three modes of operation are provided for DCEMRI.jl. First, it can be called as a command line tool using the provided script dcefit. This mode is appropriate for batch processing, or as part of shell scripts or larger analysis programs written in languages other than Julia.

The second model of operation is through the supplied MATLAB interface. Results can be saved as a MAT file, and then passed to the MATLAB function dcefit.m. Saving a MAT file is not as fast as direct parameter passing, but the data sizes in DCE MRI are typicall small enough relative to the computational complexity of the problem that saving and reading from disk is fast compared to the total processing time.

Finally, the preferred interface is as a direct Julia module. The DCEMRI.jl package is built as a proper Julia module. It can be loaded with the command using DCEMRI, and then inside a Julia program the provided functions can be called directly. In fact, the dcefit command-line interface does exactly this, with some intermediate command-line argument parsing. Loading the module in an interactive Julia session will exploit the precompilation to make subsequent executions faster. For example, in our testing, the in vivo demo required 47 s to run (including environment loading and writing plot files) for the first run in an interactive session, but a second complete run finished in 8.5 s. This is extremely advantageous for iterative development and batch processing, when the analysis might need to be run many times.

Results and Discussion

Validation: QIBA phantom data

The first validation set was performed on the QIBA version 6 Standard Tofts-Kety phantom.11 We installed DCEMRI.jl on a 2.4 GHz Intel Xeon E5-2665 workstation running Ubuntu 14.04.1 LTS (GNU/Linux 3.8.0-30-generic) using Julia version 0.3.1 (commit c03f413). Eight CPU workers were used. For the noise-free case with 30 voxels × 1,321 time points, fitting progressed at 5.2 voxels s−1, requiring 5.8 s total; for the noisy data with 3,000 voxels, the fitting rate was 5.7 voxels s−1 and required 525 s total.

In the noise-free case, the recovered parameters matched the true values to within an RMS error of 0.419% for Ktrans and 0.126% for ve. The maximum error in the fitted parameters was 2.17% for Ktrans and 0.570% for ve. The concordance correlation coefficients (CCCs) were >0.999 for Ktrans and ve. The fits with the largest error relative to the true value occurred in the regions with the lowest ve and the highest Ktrans.

In the noisy case, using σ = 0.2, the recovered parameters agreed with the true parameters to within an RMS error of 21.5% for Ktrans and 16.1% for ve. The CCCs were 0.866 for Ktrans and 0.871 for ve. In contrast to the noise-free case, the lowest Ktrans and ve values had the largest relative error. The resulting parameter maps and associated errors are shown in Figs. 2 and 3.

Figure 2 Fitting Standard Tofts-Kety model parameters to the QIBA v6 noise-free phantom.

The RMS error was 0.419% for Ktrans (A, C) and 0.126% for ve (B, D). CCCs were >0.999 for both parameters.

Figure 3 Fitting Standard Tofts-Kety parameters to the QIBA v6 phantom with sigma = 0.2 noise added.

The RMS error was 21.5% for Ktrans (A, C) and 16.1% for ve (B, D); CCCs were 0.866, and 0.871, respectively.

The second validation set was performed on the QIBA version 4 Extended Tofts-Kety phantom. For this example, we removed the regions with Ktrans = 0 min−1 from the phantom, since no transfer from the blood to the tissue violates the two-compartment model assumptions and precludes any estimation of ve. The same software and hardware setup was used as in Validation 1. Again eight CPU workers were used. For the noise-free case with 90 regions and 661 time points, fitting progressed at 20.8 voxels s−1, requiring 4.3 s total; for the noisy data with 9,000 voxels, the fitting rate was 19.7 voxels s−1 and required 456 s total.

In the noise-free case, the recovered parameters matched the known truths to within an RMS error of 6.97% for Ktrans, 18.0% for ve, and 23.8% for vp. The CCCs for both parameters were >0.999 for Ktrans, 0.890 for ve, and >0.999 for vp. The fits with the largest error relative to the true values occurred in the regions with the lowest Ktrans and, to a lesser extent, lowest vp.

In the noisy case, using σ = 0.2, the recovered parameters agreed with the true parameters to within an RMS error of 11.3% for Ktrans, 18.2% for ve, and 12.7% for vp. The CCCs were 0.974 for Ktrans, 0.703 for ve, and 0.972 for vp. Against the fits with the largest relative error occurred in regions of interest with the lowest Ktrans and vp. The resulting parameter maps and associated errors are shown in Figs. 4 and 5.

Figure 4 Fitting Extended Tofts-Kety model parameters to the noise-free QIBA v4 phantom.

The RMS error was 6.97% for Ktrans (A, D), 18.0% for ve (B, E), and 23.8% for vp (C, F); CCCs were >0.999, 0.890, and >0.999, respectively.

Figure 5 Fitting Extended Tofts-Kety parameters to the QIBA v4 phantom with sigma = 0.2 noise added.

The RMS error was 11.3% for Ktrans (A, D), 18.2% for ve (B, E), and 12.7% for vp (C, F); CCCs were 0.974, 0.703, and 0.972, respectively.

Several factors likely contribute to the accuracy of retrieving perfusion parameters from the QIBA phantom data. Most importantly, the Jacobian of the Tofts-Kety model includes terms of the form (5) ∂Ctt∂Ktrans=∫0tCpsexpKtransves−t1+Ktrans2ves−tds,

(6) ∂Ctt∂ve=−Ktrans2ve2∫0tCpsexpKtransves−ts−tds,

and (7) ∂Ctt∂vp=Cpt.

With such strong dependences on Ktrans and ve, the Ktrans and ve columns of the Jacobian may become ill-conditioned when Ktrans or ve take on extreme values, leading to a loss of numerical precision. For example, in the Standard Tofts-Kety model, the difference in dependence of the terms on ve can cause ill-conditioning when ve is close to zero, regardless of Ktrans, since both columns depend on Ktrans in similar ways.

This hypothesis is strengthened by observing that the largest error in Figs. 2 and 3 are when ve = 0.01 and is roughly independent of Ktrans. We also note that for the model assumptions to be valid ve must be non-zero. Thus the error increases as the parameters get closer to violating the model assumptions.

For the Extended Tofts-Kety model, the situation changes because the Jacobian has a different term, vp, that is independent of Ktrans and ve. Because of this, Ktrans alone can now cause ill-conditioning. Figures 4 and 5 are consistent with these limits. The largest error occurs for Ktrans = 0.01 min−1 and ve = 0.5. The error in the fits can also be said to be largest when kep is small. This suggests that the numerical precision of the fits should be much lower in regions of low transfer and high extravascular, extracellular volume, such as the central, necrotic regions of some tumors.

Quantitatively, the two validation data sets recovered the parameters extremely accurately when Ktrans ≥ 0.05 min−1. Because of this, we recommend caution when including voxels where Ktrans < 0.05 in analyses.

As an aside, a very slight underestimation Ktrans and ve is apparent in the pinkish tint of the error maps of Fig. 2, and a slight overestimation can be seen in the greenish tint of the error maps in Fig. 4. Neither of these effects is large when compared to the other fitting issues, however. We have no explanation for this, but note it here as a point of curiosity.

At any time, the QIBA validations may be run automatically by the user with the command validate(). This allows subsequent software versions to be validated, or the results of this section to be reproduced.

Application: in vivo breast DCE-MRI

The third validation was not a test of accuracy of parameter recovery, but rather a proof of concept for in vivo applications. In vivo data is more hetereogeneous and subject to measurement error and voxel averaging, so not all measured voxels may follow the Standard or Extended Tofts-Kety model.

The same hardware and software setup was used as in Validations 1 and 2. A standard Tofts-Kety model was used because of its robustness to noise and for simplicity of exposition here. DCEMRI.jl selected 18,327 voxels as containing significant signal and created R1 and S0 maps in 2.9 s, for a processing rate of 6,365 voxels s−1. Of the 18,327 voxels selected for R1 fitting, 6,774 were computed to have a signal enhancement ratio of 2.0 or more and were selected for DCE model fitting. Fitting required 0.9 s, for a rate of 7,815 voxels s−1. The resulting maps are shown in Fig. 6.

Figure 6 In vivo example.

The computed R1 relaxation rate (A), signal enhancement ratio SER (B), maximum CA tissue concentration Ct (C), the Standard Tofts-Kety parameters Ktrans (D) and ve (E), and the fit residual (F). Only voxels with SER >2.0 were fit. Of the 36,864 voxels in the image, only 7,815 were selected for parameter fitting.

The general features of the computed maps for the in vivo are consistent with expected results. The R1 relaxation rate is lower in the tumor than in the fatty tissue and is similar to that in the fibroglandular tissue. The CA concentration is generally highest in the tumor and in vascular-like structures. The signal enhancement ratio is highest in the tumor, apart from some garbage results posterior of the chess wall due to breathing and cardiac motion. Finally, the derived values of Ktrans, ve, and vp through the tumor are consistent with typical tumor values and spatially consistent with neighboring voxels.

Run time

We have collected the run times and number of time points of each of the two cases for each of the two validations along with the same for the in vivo example in Fig. 7. We wanted to determine the scaling of run time of DCEMRI.jl with problem size so that better comparisons with other packages can be made. We hypothesized that the run time would be dominated by the matrix operations in the Levenberg–Marquardt routine. Under this hypothesis, we assumed that a polynomial scaling of the run time might occur. Since matrix multiplication can scale as poorly as O(N3), we tested low-order polynomials in N and logN, but we found poor fits when a zero intercept was required. A power-law fit was found to fit better than polynomials, and we found that the best fit power law for the run time in seconds per voxel was trun(N) = 2.2 × 10−7N1.9, where N in the number of time points per voxel.

Figure 7 Time required to fit a single voxel as a function of the number of time points in the Ct curve.

Comparison to other packages

We ran the two most similar packages—DCE@urLAB and dcemriS4—on a 2.4 GHz Intel Xeon E5-2665 workstation running Ubuntu 14.04.1 LTS (GNU/Linux 3.8.0-30-generic) using Julia version 0.3.1 (commit c03f413). Eight CPU workers were used.

DCE@urLAB by Ortuño et al. (2013) found comparable errors in fitting to this work. While they didn’t state quantitative error measurements, their Figs. 7 and 9 were similar in character to Figs. 2–5 here. They also stated run times of 20 s to fit 1,024 voxels and 40 dynamic frames and 5 min to fit for 16,384 voxels and 40 dynamic frames, or roughly 19 ms per pixel. According to our run time analysis, DCEMRI.jl using four CPU cores would require only 0.80 ms per pixel which is 24 × faster.

Also, DCE@urLAB contains around 20,000 lines of code, while DCEMRI.jl contains only around 1,000 lines of code, and 500 of those are devoted to phantom validation and plotting. DCE@urLAB does contain more models, however, so adding additional models to DCEMRI.jl will require more code, but only on the order of tens of lines, not thousands.

Validation results for dcemriS4 have not been published, so we cannot compare its accuracy to DCEMRI.jl, but in our own testing, we found that dcemriS4 required roughly 10 s on average to fit the Extended Tofts-Kety model to the tissue curves derived from the breast data set, while DCEMRI.jl required 0.9 s. This suggests that DCEMRI.jl may be ∼10 × faster than dcemriS4.

Conclusions

We have demonstrated an open source, free, and highly portable solution to DCE-MRI analysis that achieves similar accuracy of derived parameters, eschews needless complexity, and is 10–20× faster than comparable solutions. Many improvements are possible for DCEMRI.jl. First, more models can be added as long as they can be validated. Many of the existing packages include more than just the Tofts-Kety models, and DCEMRI.jl is written such that the model to be fit is completely independent of the fitting code itself. Thus adding new models is trivial. Second, upcoming improvements to the Julia language will bring even more speed. The planned feature of module load caching should speed up loading modules in Julia, which is currently one of the slowest parts of DCEMRI.jl. A major overhaul of plotting in Julia is also in progress which should improve the speed and quality of plotting. Finally, improvements in the low-level code translation to better optimize vector arithmetic on modern CPUs are in the works. As these changes are implemented, users can stay up to date simply by updating to the latest Julia stable release and then running Pkg.update() in the Julia environment. This command will pull the latest commits from the DCEMRI.jl git archive and patch the local copy of the module.

We thank Professor Mark D. Does of the Vanderbilt University Department of Biomedical Engineering for helpful discussions.

Additional Information and Declarations

Competing Interests

Author Contributions

Human Ethics

Data Deposition

1 http://www2.die.upm.es/im/archives/DCEurLAB/

2 http://ikrsrv1.medma.uni-heidelberg.de/redmine/projects/ummperfusion

3 http://thedcetool.com/

4 Mathworks, Natick, MA.

5 http://dcemri.sourceforge.net/

6 http://julialang.org/benchmarks/

7 http://github.com/davidssmith/DCEMRI.jl

8 https://www.rsna.org/QIBA.aspx

9 Philips Healthcare, Best, Netherlands.

10 Vanderbilt Ingram Cancer Center Institutional Review Board Protocol BRE 0588: MRI Evaluation of Breast Tumor Growth and Treatment Response.

11 https://dblab.duhs.duke.edu/modules/QIBAcontent/index.php?id=1

The authors declare there are no competing interests.

David S. Smith conceived and designed the experiments, performed the experiments, analyzed the data, contributed reagents/materials/analysis tools, wrote the paper, prepared figures and/or tables, reviewed drafts of the paper.

Xia Li analyzed the data, contributed reagents/materials/analysis tools, reviewed drafts of the paper.

Lori R. Arlinghaus performed the experiments, analyzed the data, contributed reagents/materials/analysis tools, reviewed drafts of the paper, collected in vivo data.

Thomas E. Yankeelov contributed reagents/materials/analysis tools, reviewed drafts of the paper, IRB approval; collected in vivo data.

E. Brian Welch performed the experiments, analyzed the data, contributed reagents/materials/analysis tools, reviewed drafts of the paper.

The following information was supplied relating to ethical approvals (i.e., approving body and any reference numbers):

1. Vanderbilt Ingram Cancer Center Institutional Review Board.

2. Protocol BRE 0588: MRI Evaluation Of Breast Tumor Growth And Treatment Response.

The following information was supplied regarding the deposition of related data:

http://github.com/davidssmith/DCEMRI.jl

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
