# Peer review of "DCEMRI.jl: a fast, validated, open source toolkit for dynamic contrast enhanced MRI analysis"

_PeerJ, doi:10.7717/peerj.909_

## Round 0.1 · original submission · Major Revisions

· Academic Editor

Major Revisions

Dear authors,

See reviewer 2's comments that "This is a highly technical paper that may not be suitable to our journal. Readers would be interested in using the software, but not in how it is working. "

However if you can improve your paper to add a bit more clinical value that would be great.

·

Basic reporting

+ professional standard of the manuscript
+ clearly structured manuscript
+ relevant figures
+

Experimental design

+ clearly defined study protocol
+ high technical standard

Validity of the findings

+ statistically sound data
+ the conclusion is reproducible
+ the presented DCE tool-kit seems to be highly-developed and the conclusion is reproducible

Additional comments

+ the presented software seems to be an alternative to the MR-system implemented image-processing software, which is (license!) very expensive.
+ the presented example of mamma-tumor-perfusion is very impressive. Is the production of such a perfusion map a highly time-consuming process? How about archiving these maps in a PACS system?

·

Basic reporting

See general comments

Experimental design

See general comments

Validity of the findings

See general comments

Additional comments

SUMMARY: A software package is described that analyzes Dynamic Contrast Enhanced MRI data for biomedical imaging purposes. Software exists, but is not open source, slow, properly validated, complex and not flexible. A new software is introduced that uses Julia, a novel programing language, that is claimed to be simple and communicates well with Matlab, a closed source package often used by researchers. They implemented a couple of DCEMRI models and implement fitting routines in a straightforward fashion. the result is quantitative maps that can be used to infer tissue properties. The method is explained with many equations. The software is validated using the recent DCEMRI digital phantom data made available by the QIBA group from RSNA. The software is shown to be QIBA compliant. Furthermore a a single breast MR from one breast is used as an example. It claims to be open source, portable, faster.

Comment.

1. This is a highly technical paper that may not be suitable to this journal. Readers would be interested in using the software, but not in how it is working.
2. The open source model for this software is not clear. It is now living in the GitHub maintained by the first author. This is not going to work in the future. How can one be sure that the software remains QIBA compliant? How about problem reporting? Either one should setup a community or hook up to an existing community. See e.g. VTK/ITK from Kitware, or Slicer. Can this journal also host software?
3. The claim of speed is not exactly clear. The tests lack information about hardware, so it is unclear how results can be compared.
4. Speed is even more strongly dependent on the implementation of DCE analysis. The authors chose straightforward implementation of DCE analysis methods, like deconvolution. This is notoriously slow. Other methods are available in literature (e.g. matrix based) that are much faster. Probably requires a decision to go for speed or ''simplicity''.
5. The T1 relaxation is estimated by assuming a known MR signal model (e.q. 1). This is known to fail for the regular clinical sequences. They just do no fit these standard gradietn echo modesl. How are the parameters derived? Are there other signals models? How should these parameters be derived?
6. The method assumes the availability of a mask. How is that created? Is that available as well?
7. The clinical breast MR example is highly unrealistic. It is an MR exam of a single breast and extremely slow. Most likely because the authors selected a sequence in which the signal model does behave as Eq. 1. As I commented in 5, that is not realistic. Regular breast MR should be at least 1 mm in resolution, span both breasts and run under 90 seconds. In that case you will have to use faster sequences, that definitely not fit the Eq.1 model.
8. The breast MR is a mere example and not in any way a validation of the method. There is no reference standard, no comparison.

---

## Round 0.2 · accepted · Accept

· Academic Editor

Accept

Well done review - the paper has become very interesting !!

·

Basic reporting

No comments

Experimental design

No comments

Validity of the findings

hardly verifiable...

Additional comments

the point of view of a radiologist is very simple, can I trust the results and how can I implement the maps in my daily routine workflow...
usually the radiological institutions have to pay good money for the licenses of software modules run by the equipment manufacturers.
Therefore an open source freeware for DCE MR mapping sounds very interesting, nevertheless we should compare your results with the standard software....

·

Basic reporting

none

Experimental design

none

Validity of the findings

none

Additional comments

The rebuttal was to the point and impressive in its honesty.
The modifications to the manuscript addressed all my comments.
I am happy to accept the manuscript as is for this journal.